# An FPGA Based Tracking Implementation for Parkinson’s Patients

**DOI:** 10.3390/s20113189

**Published:** 2020-06-04

**Authors:** Giuseppe Conti, Marcos Quintana, Pedro Malagón, David Jiménez

**Affiliations:** 1Visual Telecommunications Applications Group, Universidad Politécnica de Madrid, 28040 Madrid, Spain; djb@gatv.ssr.upm.es; 2Everis ADS, Camino Fuente de la Mora, 1, 28050 Madrid, Spain; marcos.quintana.gonzalez@everis.com; 3Integrated Systems Lab, Universidad Politécnica de Madrid, ETSI Telecomunicación, 28040 Madrid, Spain; malagon@die.upm.es; 4Center for Computational Simulation, Campus de Montegancedo, Universidad Politécnica de Madrid, 28660 Madrid, Spain

**Keywords:** image processing, image analysis, human detection, patient privacy, tracking, GMM, background subtraction, FPGA, MoG

## Abstract

This paper presents a study on the optimization of the tracking system designed for patients with Parkinson’s disease tested at a day hospital center. The work performed significantly improves the efficiency of the computer vision based system in terms of energy consumption and hardware requirements. More specifically, it optimizes the performances of the background subtraction by segmenting every frame previously characterized by a Gaussian mixture model (GMM). This module is the most demanding part in terms of computation resources, and therefore, this paper proposes a method for its implementation by means of a low-cost development board based on Zynq XC7Z020 SoC (system on chip). The platform used is the ZedBoard, which combines an ARM Processor unit and a FPGA. It achieves real-time performance and low power consumption while performing the target request accurately. The results and achievements of this study, validated in real medical settings, are discussed and analyzed within.

## 1. Introduction

Nowadays human pedestrian tracking and object detection systems are taking on an important role in real life applications. These systems attract considerable interest in the industrial and scientific communities, and a lot of methods have been proposed and developed. Moreover, computer vision is constantly growing, and its deployment is already reaching embedded systems. New customizable hardware (Hw) solutions, operating systems (OS) and software (Sw) applications have already been developed for research purposes, concluding in robust prototypes that have derived in ready-to-market solutions. These systems are integrated in a single chip and embedded on a board. These emerging technologies known as FPGA–SoC (field programmable gate array–system on chip) and FPGA-SoM (field programmable gate array–system on module) consist of a chip that encapsulates both FPGA and a processor (such as an ARM core) that shares the entire system resources (RAM memory, connections, peripherals, SD storage, etc). These technologies, already adopted within cameras for sensor acquisition and coding stages [1,2,3], allow us to observe real time performance for complex computer vision systems capable of providing flexibility, high precision and low power consumption. There are a wide scope of applications for this technology, such as advanced driver assistance systems (ADAS), industrial inspection applications, video surveillance, traffic surveillance, pedestrian detection and tracking, patient monitoring, biomedical devices, etc. Biomedical health and life sciences represent special sectors where the use of SoC and SoM is becoming popular. Thanks to their features and flexibility, these embedded platforms provide excellent options to improve healthcare and biomedical-related processes, such as the device described in [4].

This is possible due to the advantage of using a SoC or a SoM, which allows for the combination of existing software (SW) libraries and hardware (HW) acceleration in a single compact device. As a consequence, this technology makes it possible to reduce both the size and power consumption of traditional medical and biomedical equipment.

A comparison between CPUs, GPUs and FPGAs is presented in [5] and also shows the advantages of using FPGAs instead of GPUs or CPUs. The work also compares three different algorithms GMM (Gaussian mixture model), ViBE (visual background extractor) and PBAS (pixel-based adaptive segmenter), which are all excellent options for developing the tracking systems. An interesting part of the study focuses on the precision and the HW accuracy, including a complete comparison based on those metrics.

A new method for moving object tracking, which improves the traditional KLT (Kanade–Lucas–Tomasi) algorithm, is presented in [6]. The improved KLT algorithm proposed by the authors enhances the extraction feature by introducing the motion information to filter out the static corner points. Additionally, a frame processing order strategy is adopted to take advantage of FPGA parallelism and reduce the execution time. The authors’ work is remarkable, and it has been compared with our proposal in Section 7.

In [7] a tracking system for amphibious spherical robots was implemented. The robots in this system use a GMM to detect a moving target entering the field of view. The authors achieved good results employing 8-bit gray scale frames, and an internal image resolution of 160 × 120. They used a OV7670 CMOS VGA-specific camera with PMOD interface with I2C protocol and a parallel specific interface to communicate with FPGA. The authors also studied the precision and discussed their implementation drawbacks.

A different approach for background subtraction is presented in [8], where the authors present an algorithm that uses the hue component of HSV color space [9] and the gray-scale pixels’ intensities to implement an efficient background subtraction algorithm for real time applications. The authors of this paper made a special effort to to analyze the performance metrics and the quality of the foreground mask.

A surveillance Zynq XC7Z020 FPGA board (ZedBoard) based tracking system is presented in [10]. The authors used an OV7670 dedicated camera, controlled by I2C, directly connected to the ZedBoard PMOD input bus. The resolution of the processed signal is 640 × 480, with an 8-bit grayscale. The method employed to complete the segmentation between background and foreground regions is a pixel subtraction between a stored frame (background) and an input frame.

Another relevant method is presented in [11], where a Siamese Neural Network is implemented focusing on the mobile devices. The authors reduced the 32-bit floating point and activation parameters into 8-bit keeping a good precision. They also reduced the inference time of the network using pruning and quantization techniques. The tests are done using the VOT2015 dataset, whose sequences are not constant in terms of resolution. Finally, the system is compared with VOT2015 dataset testing results of 62 state-of-the-art trackers, and their FPGA implementation achieves a robust energy-efficient and area-concise solution.

The solution presented in [12] introduces a moving object detection and tracking system based on dynamic background difference and Kalman filtering. Wenchao Liu et al. improved the dynamic background difference combining the time averaging method and the recursive filter for the real-time background image updating. The processing system combines the HW/SW resources to archive real-time performances.

Finally, the last relevant study to mention is based on optical flow estimation and a Gaussian mixture model [13]. The authors proposed a novel algorithm focused on environments wherein the camera is not static; the study was implemented and tested on a Xilinx Virtex-5 FPGA. Jaechan Cho et al. compared their work with previous research, and found that they had achieved good performances in terms of speed, precision and detection accuracy thanks largely to the contribution of the optimized hardware structure and the background compensation.

The previous study is considered to be the model for the improved system presented within this paper. The Gaussian mixture model is adopted to segment the frames extracted from the video sequence and extract the regions which with a high probability belong to moving objects (humans in our case). Jaechan Cho et al. proved that this algorithm is suitable to be embedded on a FPGA device, and therefore, to ease the deployment of the system and reduce its power consumption. In our case we have improved their results by optimizing the HW architecture by including a temporal buffer, reducing the number of operations and improving the communications among the different modules. In order to complement the proposed HW implementation and test a complete solution for trajectory estimation, a Kalman Filter [12] has been included in the system as well. This filter has been widely used in related literature to propose a temporal model and predict the current location of the object based on the previous position and the current estimation during the detection step.

### Contributions and Structure of the Paper

The main contribution of this paper is the design of an embedded system that uses OpenCV Libraries to track patients with Parkinson’s disease, tested in the Asociación Parkinson Madrid (APM) (Asociación Parkinson Madrid (APM) http://www.parkinsonmadrid.org) day hospital. The system, following the specifications of the requirements, was developed using a ZedBoard, a FPGA-SoC based board that enables us to accelerate the algorithm. This board selection provides the maximum flexibility to integrate the solution, susceptible to changes during the life of the project. A special effort of considering the patients privacy has also been done, because the system cannot store the streamed video. To achieve this result, a novel architecture was designed with a special combination of HW/SW capabilities, where a mixture of Gaussian (MoG) and a Kalman filter trackers were employed, optimizing the background subtraction to be executed on the FPGA. This marks an important state-of-the-art advance from previous studies, by ensuring the complete interaction and management of the video stream. While other studies focus on the background subtraction, or on the processing part, in this paper we consider the complete process chain, including the relevant times needed by the input and the output stages. In fact, this hybrid system aims to accelerate the software solution designed for the Parkinson’s patients, taking advantage of the Zynq platform flexibility. After the computational analysis of the algorithm, the FPGA performs the background subtraction and provides its output to the CPU that implements the Kalman filtering and estimates the coordinates of the patient at every frame. This output represents the end of the complete processing chain and it is sent to a control server.

This paper is organized as follows: In Section 2, the problem statement, notation and assumptions are addressed. In Section 3, the video processing architecture and the design methodology are presented. In Section 4 the proposed algorithm is detailed. Section 5 describes the architecture implementation and integration. In Section 6 the experimental setup is presented. Moreover, in Section 7 the results of the system are illustrated. Finally, conclusions and future work are drawn in Section 8.

## 2. Problem Statement

This work has been carried out within the larger framework of the ICT4Life project [14]. ICT4Life aims to develop a complete platform with the goal to deliver an innovative service to patients affected by dementia-related diseases, mainly Parkinson’s and Alzheimer’s. By integrating video analysis capabilities into the platform, visual monitoring of patient behavior and patient progress is enabled. This type of information is useful for health professionals and both formal and informal caregivers, since it helps them to estimate the severity and evolution of the patients’ health and alert them in case of issues related to Parkinson’s, such as accidents resulting from falls. The overall system has been developed following a user-centered methodology and validated in real life scenarios, such us the one shown in Figure 1.

The platform, described in [15], consists of different sensors: visual, motion and depth, that together form a multi-sensor tracking system [16]. The types of devices vary based on the requirements analyzed for the different scenarios and the typology of patients, including:Wearable bracelets providing patients’ motion data from embedded gyroscope and accelerometer, and health information, such as heart rate and skin temperature [17].Location sensors, a binary non-intrusive sensor placed to delimit the patient’s behavior analysis zone.Depth vision cameras (Microsoft Kinect v2) for extraction of depth information to integrate deep motion analysis to monitor user activity, status and evolution.360-degree panoramic camera (Zenith), covering targeted areas such as halls and living rooms in senior centers. It provides better images to deploy the tracking algorithms and avoid occlusions.Wireless sensor network (WSN) anchors monitoring of the radio signals from patients’ wearables in a non-intrusive manner. They provide extra info to determine the trajectories derived from the camera images. [18]. Anchors are placed on the roof in the same areas the cameras are installed.

All the information coming from the sensors is integrated and fused in an adaptive multi-sensory application. The real-time collected data are synchronized and organized in this application. To fulfill ethical requirements, the system allows safe storage of information, specifically of the processed data, with a privacy-by-design approach. Figure 2 depicts the architecture and interconnection of sensors.

In accordance with the objectives of the project, the real-time requirements, and the necessary flexibility and openness were fundamental in the selection of the hardware framework to be used for the implementation. The Zedboard is a low-cost development board for integrating implementations over the Xilinx Zynq combining an FPGA and a processor on a single chip. It establishes very fast and programmable connections between the processor and the FPGA ideal for real time video processing. Furthermore, it is flexible by not imposing tight requirements on developers. More concretely, the work in this paper is focused on human pedestrian tracking using the Zenith camera (depicted in the orange dotted square of Figure 2). An important requirement is the need to track the patients while also respecting their privacy. To meet this need, the camera stream is not recorded as is. This has been achieved by the design of a system module to calculate the spatio-temporal coordinates/trajectories of the patients.

The patients with cognitive diseases such as Parkinson’s, have problems regarding mobility and often do not actively move. The trajectory data are relevant because they provide daily estimations about how much the patients are moving. This information helps the doctors to understand the patient’s status, and to modify the dosages of their medications based on these observations. To balance the aims of the video monitoring system, the context of the patients and the real-time needs, the video frame rate has been fixed at 15 fps. This selection provides enough frames to capture most actions granularly while minimizing processing costs.

The Figure 3 shows the different elements involved in the module previously described, where the camera is connected with the FPGA board, which processes the video stream and sends the data to the server. The system communicates via an Ethernet network connection.

The camera used in this project is a fish-eye camera by Vivotek; more specifications are available in [19]. There are two basic features that should be taken into account when developing a real-time video processing solution: the acquisition frame rate and the input resolution of the video. The sensory device chosen can work with different resolutions (results are provided in the following sections) and a frame rate of 15 fps. Therefore, we have assumed that to grant real-time processing each frame should be processed at around 66.67 ms or less. This requirement represents the time to meet by the algorithm.

The workflow of the global algorithm implemented for tracking is depicted in Figure 4. Here a background subtraction module has been implemented in a FPGA device. The corresponding CPU sends the frame to be analyzed to the HW device which processes the corresponding bounding boxes and returns its coordinates. If the bounding box fits the features of a person in the corresponding scenario, the extracted bounding box will feed the Kalman tracking.

## 3. Video Processor Architecture

This section describes the chosen design methodology and the proposed structure of the system as a processing chain. Following the design methodology, we first analyze the timing of a solution in a software-based implementation and include a discussion on how to accelerate it using an FPGA in order to meet the timing requirements.

### 3.1. Design Methodology

The ICT4Life consortium European research project has stated that 15 FPS is the lower limit to track Parkinson’s-affected patients. As the detection and tracking algorithm is being evolved, we require a flexible solution for modeling and prototyping (mainly in SW) with the potential for posterior editing for optimization purposes, in case the requirements are not satisfied. It is important to follow a design methodology that enables the functional and model evaluation of the algorithm via software simulation, while enabling module modifications. Last, we select an embedded system, so it can be deployed in the target location, or even integrated in the webcam.

Considering these requirements we propose processing the tracking information close to the camera, using a hybrid System-on-Chip based solution, with CPU and FPGA. The SoC board is connected to the Zenith camera through IP/TCP. Based on the latest advances, with special mention to the work of Bulat et al. [5] which compared three tracking algorithms focusing on the advantages of the use of FPGAs over GPU and CPU, the FPGA-SoC is a suitable choice for the base architecture to run and to accelerate the algorithm, and to meet the required timings and also respecting the privacy needs of Parkinson’s patients by gathering only the tracking information, without storing images or video data.

For prototyping, we selected the Zedboard, with the Xilinx Z-7020, available when the project started. The board includes a Linux OS, in order to accelerate the integration of the module in the system, as it provides mechanisms to add libraries and modules to handle IP/TCP, Web Server, multithreading, memory handling, etc. The tracking module was developed in C++ using OpenCV libraries. When required, Xilinx Vivado HLS offers the possibility of designing and modelling in C++, providing, as output, a synthesized submodule for the SoC’s FPGA. The C++ sub-modules can be both simulated and synthesized, considering some restrictions in code structure and implementation that are detailed in Section 5. A pure HW implementation using VHDL or Verilog HDL could achieve a higher level of optimization, but in this case the use of the original C++ algorithm flow and the interaction with OpenCV libraries is essential to maintain the functionality of the model, grant its verification and allow future modifications and improvements of the system.

We propose a software-based solution, divided in sub-modules according to functionality. Then, it is possible to measure the video frame rate that can be processed. In order to test the solution, some recordings have been saved as described in Section 6. If the 15 FPS requirement is not satisfied, then the sub-modules are evaluated individually. The module with the highest computing costs is the one selected to be optimized, firstly by software, and then implemented in hardware if needed.

### 3.2. Proposed Structure

In the Figure 5 the processing chain is presented, divided into 4 sub-modules. The input is a video stream processed frame by frame. The first module checks the video stream to evaluate its correctness (resolution, video components, etc.). Next, a background subtraction is performed. In the following stage, a contour finder and blob extraction algorithm prepare the data for a Kalman tracker that calculates the trajectories of the patients. Finally, the output is generated as a JSON format stream, ready for its transmission over the network.

We have first developed a SW version of the system written in C++ language and using OpenCV 3.1 libraries. The background subtractor block is implemented using MoG [20], with both gray-scale and RGB inputs, considering different resolutions. The tracking stage that follows the MoG is performed by a Kalman Appearance Tracker [21]. The target’s motion is modeled by a 2-D translation at a single scale and searched in an area around the previous target position. In the subsequent frame, the candidate windows around the predicted target position are reduced and compared to the template.

The execution time of the entire processing chain did not initially meet the requirements. Consequently, we have evaluated the computational load of each block of the algorithm, and decided to accelerate the background subtractor block in hardware. This means that the FPGA hardware implementation replaces the OpenCV software implementation to reduce the processing time, as indicated in Figure 5. It is possible to distinguish two paths: the red represents the full-software process, the slower one, while the green path represents the accelerated version implemented in the FPGA.

## 4. Background Subtractor Algorithm

The algorithms chosen for HW acceleration were MoG because of its stability for background subtraction, and a Kalman filter for the tracking because of its robustness based on a statistical model. This section is focused on the initial step (MoG) embedded on a FPGA device, whose purpose is to segment between foreground (FG) and background (BG) regions in the scene. A statistic model is built to cover the static elements in the scene, and those elements that do not fit into that distribution will be considered regions of interest (ROIs). Those ROIs that correspond to moving objects will provide the initial step towards defining the pedestrian trajectories.

Different people may enter or leave the corresponding scene dynamically. Therefore, an adaptation for those changes is required; accordingly, the solution proposed in [22] has been initially adopted. The initialization set is updated by adding new samples and discarding the old ones. At time *t*:(1)$$X^{T}=x^{t},\ldots ,x^{t-T}$$

When a new sample jumps into the scene, the initialization set $X^{T}$ is updated and $\hat{p}\left(x|X^{T},BG\right)$ is re-estimated, the probability of every pixel ($\hat{x}$) (with the prior knowledge of the history) is modeled by a mixture of *K* Gaussian distributions ($X^{T}$ ) that define the static elements of the scene (BG). However, in the recent history there could be some values that belong to the foreground (FG) objects, and this estimate is denoted by $\hat{p}\left(\bar{x}|X^{T},BG+FG\right)$. MoG is employed with M=3 components
(2)$$\hat{p}\left(\bar{x}|X^{T},BG+FG\right)=\sum_{m=1}^{M}\hat{\bar{\pi_{m}}}N\left(\bar{x};\mu^{m},{\hat{\sigma }}_{m}^{2}I\right)$$
where $\hat{\bar{\pi_{1}}},\ldots ,\hat{\bar{\pi_{M}}}$ are the estimates of the means and ${\hat{\sigma }}_{1},\ldots ,{\hat{\sigma }}_{M}$ are the estimates of the variances that describe the Gaussian components. The covariance matrixes are assumed to be diagonal and the identity matrix *I* has proper dimensions. The mixing weights denoted by $\hat{\bar{\pi_{m}}}$ are non-negative and add up to one.

Efficiency of this initial proposal was improved upon [23] by density estimation. Thus a uniform kernel is used to count the number of samples *k* from the data set $X^{T}$ that lie within the volume *V* of the kernel [23]. The volume *V* is a hypersphere with diameter *D*. The density estimate is given by:(3)$$\hat{p}\left(\bar{x}|X^{T},BG+FG\right)=\frac{1}{TV}\sum_{m=t-T}^{t}b^{\left(m\right)}K\left(\frac{\left|\right|{\bar{x}}^{m}-\bar{x}\left|\right|}{D}\right)$$
where the kernel function K(u)=1 if u<1/2 and 0 otherwise. The volume *V* of the kernel is proportional to $D_{d}$ where *d* is the dimensionality of the data. In practice *T* is large and keeping all the samples in $X^{T}$ would require excessive memory and calculating the previous equation would be too slow. It is reasonable to choose a fixed number of samples K<<T and randomly select a sample from each sub-interval T/K.

The $X^{T}$ also contains samples from the foreground. Therefore, for automatic learning a set of the corresponding indicators $b^{\left(1\right)},\ldots ,b^{\left(T\right)}$ is kept. The indicator $b^{\left(m\right)}$ has a value 0 if the sample is assigned to the foreground. The background model considers only the samples with b(m)=1 that were classified as belonging to the background. Therefore, if $\hat{p}\left(\bar{x}|X^{T},BG+FG\right)>c_{thr}$ the pixel is classified as background.

## 5. Implementation

### 5.1. Hardware/Software Interface

Vivado HLS is used to generate an intellectual property (IP) HW block. The bitstream file created represents the final result of the Xilinx tools workflow, with the HW optimized background subtraction module, and it is shown in the lower part (programmable logic) of Figure 6 where the complete system module is presented. In fact, it is possible to distinguish two zones: the processing system (CPU) zone and the programmable logic (FPGA) zone; a global description about their interaction while running the algorithm must also be noted. The Zenith camera inputs the video streaming through the Ethernet port to the CPU, which extracts the video frames and writes them to the input buffer in RAM. Once ready, the CPU communicates with the FPGA, that reads the video stream from the input buffer, processes the background subtractor and writes the result in the RAM out buffer. The CPU receives a signal from the FPGA specifying that the processing is done, and reads the output buffer. In the final stage, the CPU reads the output buffer and introduces the output stream from the last part of the algorithm, where the Kalman tracker extracts the people coordinates and trajectories. The JSON streaming then is prepared with the processed data and outputs it using the Ethernet interface from the CPU to the control server.

The communication between the CPU and the FPGA is done by using the AXI interface [24]. Specifically, there are three types of AXI buses:AXI master is employed for memory-mapped interfaces and allows high throughput bursts of up to 256 data transfer cycles with just a single address phase.AXI lite is a light-weight, single transaction memory-mapped interface, suitable for control signal.AXI stream that allows unlimited data burst size suitable for video streaming flows.

The FPGA internal structure showed in the programmable logic part of the Figure 6, includes two additional blocks, the AXI video direct memory access (AXI VDMA) [25] cores, which are a Xilinx IP that provide high-bandwidth direct memory access between the memory RAM and the background subtractor module. A functional description of the FPGA is the following: On one side, all the AXI lite signals (depicted in yellow in Figure 6) are the control signal bus communication with the CPU; the AXI stream (depicted in green in Figure 6) is the bus used for the video streaming; and the AXI master (depicted in red in Figure 6) is used to interchange the vector data (*modeUsed, mean, variance and weight*). The background subtractor module, which uses the three interfaces, receives the video input images from the AXI VDMA and the vector data from the AXI master bus. Once the transfer is completed, the processing starts. When the module completes the background subtraction, the output is performed using the AXI streaming for the output images, and the AXI master for the vectors. On the other side, the CPU communicates with the FPGA through two ports: the master general purpose port (M_AXI_GP depicted in purple in Figure 6) that is used to control the FPGA in every internal stage over the AXI lite interface, and the slave high performance ports (S_AXI_HP depicted in blue in Figure 6) that are used to perform the data read/write (R/W) operations on the DDR controller. The S_AXI_HP port communicates with the FPGA internal AXI stream and AXI master interfaces and permits to the FPGA the direct access to the video input/output buffers and to the vectors buffer zone.

### 5.2. Operating System

We will now provide more details regarding the different strategies applied and the modules of the systems required to run the algorithm. The OS chosen to run on the ZedBoard is Petalinux, a customizable distribution based on Yocto that allows to compile personalized versions of Linux, perfectly adapted to the HW used and including only the software packages needed. An important customization is the modification of the OS kernel, because the buffer area used to interchange data with the FPGA needs a free zone to avoid conflict with the other memory zone. For that reason the OS kernel is recompiled to use only a part of the memory, while the other is left free.

In Figure 7, it is possible to see that the RAM memory has been divided into two main parts: the OS zone, where all the Linux processes, variables and the OpenCV data are stored; and the free zone, where all the memory buffers, which contain the I/O video frames and the vectors (*modeUsed, mean, variance and weight*), are stored. The OS Kernel is also compiled to enable *userspace* I/O (UIO) driver, a method that allows it to communicate with system peripherals using the interface /dev/uioX found in the OS folder. The FPGA is addressed as an OS peripheral, and it is possible to interchange control signals and read/write the memory buffers thanks to the UIO driver, directly from the same code that implements the algorithm. A driver library has been created, allowing the complete interaction with the FPGA to the algorithm that runs on the CPU. An example of the driver communication functionality can be seen in the following scenario: the CPU receives the video sequence, writes it on the input memory buffers in the free zone and launches the background subtractor. The FPGA reads the input memory buffers, processes the data, writes the output and asserts a done signal. The CPU reads the output data from output memory buffer and provides it to the following part of the algorithm that performs the tracking to final provide the JSON data.

### 5.3. Model Modification for Acceleration

Even if Vivado HLS decides to accelerate the C++ algorithm, its code would need to be modified. This is because some C++ features are not supported, and the HDL generator requires specific software structures to translate the algorithm into a HW solution, which takes advantage of the FPGA structure and internal resource utilization. Modifying the C++ code, and creating a test bench to simulate the algorithm conditions before and after the background subtraction would provide Vivado HLS the correct translation in HDL language and proper HW generation. We enumerate and describe in the section below the main modifications applied to the C++ model:

**Remove dynamic data structures:** the use of dynamically allocated memory is not supported by Vivado HLS. The original algorithm has dynamic memory allocation with C++ pointers and OpenCV Mat type, which is the standard type used to store images. The data structure employed by MoG background subtractor in OpenCV library is composed by an input RGB image, an output grayscale image and another dynamic image that contains the algorithm data calculations with the MoG operators (mean, weight, variance and used modes). The memory elements have to be allocated statically, as variables, in the code. The structure changes applied are the following:From input image dynamic Mat to input image static matrix;From output image dynamic Mat to output image static matrix;From MoG dynamic operators to four static vectors that contain the MoG operators.

**Replace unsupported C++ features:** The original model uses *parallel_for_* loop [26], a useful framework to parallelize the code to be suitable for running and take advantage of CPU multithreading, which enables one to subdivide the operations in tasks and distribute them to each CPU core. As the FPGA internal architecture is different, the *parallel_for_* loop is not suitable to be ported on this different structure. For this reason the MoG algorithm from OpenCV library code is modified using the normal *for* and a series of adjustments to implement the algorithm in a suitable way to be supported by Vivado HLS.

**Memory usage reduction:** The FPGA has limited internal memory resources (4.9 Mbits of Block RAM (BRAM) considering the SoC XC7Z020) and the use of external DDR memory inside the hardware accelerated model would drastically reduce the throughput. Therefore, the adequate selection of data types is very important, as depending on the type of variable used by the algorithm, the memory usage is different. A data variable analysis is done to determine the data type and the range of values that are used by the algorithm. The main idea is to reduce memory utilization maintaining a correct representation of the values.

Table 1 shows the changes applied to the data. While Image In and Image Out maintain the same format, as well as *modeUsed* vector, the main change that improves the internal memory usage is applied to *Variance*, *Weight* and *Mean* vectors. These three vectors store the highest amount of data. The change from float to fixed allows to reduce the three vectors to half size, introducing precision errors due to the truncated and rounded representation of the values. This is evaluated in the Results section.

**Replace divisions with multiplications:** the factor under analysis is the learning rate of the algorithm $\alpha_{T}$ that is calculated inside the FPGA with the following division
(4)$$k=\alpha^{T}/weight$$

After testing the evolution and convergence of this factor, it has the same magnitude order of $\alpha^{T}$ and can be approximated in the following way:(5)$$k=\alpha^{T}+weight*m_{k}$$
$$m_{k}=\left\{\begin{array}{cc} 0.000288, & \mathrm{if}\;\alpha^{T}<0.025\\ 0.003221, & \mathrm{if}\;\alpha^{T}>0.025 \end{array}\right.$$
where $m_{k}$ is a constant found through empirical observation of $\alpha^{T}$ and the weight evolution. It is important to note that the approximation of the variable change from float to fixed commented on in the Table 1, produced a multiplication by zero that was not correct. For this reason it has been replaced by the addition of weight*mk. Taking into account the fixed type, the algorithm Equations (Equation 4) and (Equation 5) have a similar convergence that depends more on $\alpha_{T}$ because weight assumes values between 0 and 1, closer to one so long as the iterations of the algorithm are increased accordingly.

**Remove shadow detection:** the original MoG algorithm includes a shadows detection, which internally contains more loops and divisions. In our study, to improve the algorithm speed, this feature has been disabled.

**Duplicate variables:** Vivado HLS creates a faster HW pipelined or parallel implementations when operations are simple and the source variables are duplicated so as to be accessed as individual processes. Therefore, we make copies of variables and split the formulas performing only sums or multiplications. For example, we split this expression


 dist2 = dData[0]*dData[0] + dData[1]*dData[1] + dData[2]*dData[2];



into these

dData0a = dData0b = dData[0];

dData1a = dData1b = dData[1];

dData2a = dData2b = dData[2];

mul0 = dData0a * dData0b;

mul1 = dData1a * dData1b;

mul2 = dData2a * dData2b;

dist2 = mul0 + mul1 + mul2;



This change, even if it looks like a larger code, makes it possible to improve the parallelization, because it creates copies of the original data, avoiding collisions with other processes that need to read or write the same data. These copies made separate multiplications and are stored in different variables (registers), which are summed to obtain the final result. Vivado HLS does not translate a SW variable in C++ into a memory position, as it is typically done by compilers. It might be considered an internal signal, or translated into an internal register to perform a pipelined hardware implementation. The modification of the different formulae allows Vivado HLS to give the right priority to the creation of the final result, generating an optimized hardware structure that multiplies and sums the data. This improves the parallelization process because it allows the FPGA to read, write and process more data at the same time, since the original data are read and written in the new result registers that can achieve the results in the next clock cycles in an independent way, while the original input resources are available for other purposes or could be deleted.

**Vivado HLS pragma directives:** The last optimizations applied to the algorithm are the Vivado HLS pragma, a set of directives that makes it possible to accelerate FPGA internal loops using pipelines, improve memory resource distributions, reduce latency, improve throughput performance, etc. For more in depth information it is possible to consult the Xilinx guide [27].

### 5.4. Hardware/Software Integration

Once the driver and the algorithm have been implemented, and are ready to run on the ZedBoard, the testing and validation stages are required. The first analysis performed is on the data integrity, which consists of debugging and analysis to check if the data exchanged between the CPU and the FPGA on the free zone are correct, and if it is desired to implement the tracking in the next stage. This is performed by running a short test sequence to save images and log files that contain the values of the corresponding variables. In order to generate a reference, initially the algorithm runs only on the CPU. During the verification stage, the same test sequence is launched on the Zynq system, storing debug data from the memory buffers in the free zone, before and after the execution of background subtraction on the FPGA. The data (images and log file) are compared with the reference to check the data coherency, and to determine if the data transfer between the CPU and the FPGA has been successfully performed. If the data integrity analysis is satisfactory, the next step is to test the time analysis to evaluate the performance. All the execution times are saved to a log file, to observe and investigate any eventual losses in performance. The first bottleneck that emerges from the time analysis is the detriment to RAM speed. Unfortunately, the DDR3 memory of the ZedBoard is slow because the interface speed is governed by a 533 MHz clock, allowing data transfers up to 1066 Mbs. The FPGA internal memory resources are limited, and the amount of data that can be processed is around 15,000 RGB pixels. This means that to process an image of 640 × 480 pixels, it needs to divide it into around twenty sub-images and process them part by part. The quantity of data to transfer also requires analysis, because it could slow down the entire data (vectors and in/out sub images) transfer process, especially when this copy should be repeated several times (e.g., 20 times for 640 × 480 resolution).

Upon analysis of how the data is used in MoG, around 85% of the memory is used by the vectors that are holding all changes between the various frames. The strategy used in this study, to optimize the data copies, is to create a special buffering system where a number of frames is stored in a buffer and it is processed part by part from frame buffer 1 to *n* transferring the vectors information only at the beginning and at the end of processing. Figure 8 depicts the proposed method. The number in yellow indicates the chronology of the operations described as follows:Read vector data from RAM.Read image in part 1(2…n) from input RAM buffer and process background Subtractor, updating FPGA internal vectors’ data.Write output image part 1(2…n) to the RAM output buffer.After processing the N frames part 1(2…n), write the updated information the vectors RAM buffer.

Steps 2 and 3 are repeated *n* times until part 1 has been processed, and then continues with part 2,3,…n until the *n* frames in the buffer have been processed. The system is then ready to process the ensuing frames of the stream. This strategy improves the algorithm execution time because it drastically reduces the amount of information to copy from/to RAM. The next bottleneck to address arises in the CPU and the time that it is waiting during the data copy from/to RAM. The Zynq 7Z020 has a dual-core ARM Cortex-A9 CPU at 667 MHz, and it is possible to use C++ multi threading algorithm implementation to reduce the bottleneck represented from the data copy waiting time.

Figure 9 represents the processing strategy adopted by the multi-threading algorithm using a ping-pong buffer pattern. The frame buffer in RAM has been divided into two zones, allowing the new version of the system to copy new input data while the FPGA is processing the current frames. There are two synchronized threads in the system: thread_cp and thread_ps. First, thread_cp copies the input frames in the zone 0 input buffer. Then, thread_ps controls the FPGA to elaborate the data on zone 0 output buffer, while thread_cp is copying the following frames in the zone 1 input buffer. When thread_ps finishes in zone 0, it starts again in zone 1, while thread_cp is saving the output data and preparing the new input frames on the zone 0 input and output buffers. The subsequent executions, thread_cp and thread_ps alternate the region of execution in a sequential way processing the data. Optimizing the multi-threading algorithm allows the CPU to improve the processing time by coordinating the Zynq resources and reducing the waiting times between the various stages.

## 6. Experimental Setup

In this section, the experiment conditions are specified. For reference, the tests have been launched using a personal computer, an embedded system and a SoC using only the CPU. The specifications of the three systems are detailed below:Personal computer with an Intel i7-4790 CPU and 8GB of RAM;Raspberry Pi 3 with a 64 bit quad-core ARM Cortex A53 (ARMv8) and 1 GB of RAM [28];ZedBoard with SoC XC7Z020 (that includes Dual-Core ARM Cortex-A9 32 bit CPU and an Artix-7 FPGA) and 512 MB of RAM [29].

The experiment was performed by processing some video sequences using the three architectures, and focusing on the execution times. The times were measured at two critical points of the chain: the background subtractor and the Kalman Tracker. The results of the test are shown in Table 2.

Most of the video sequences used for the experiment are recordings of Parkinson’s patients at the Asociación Parkinson Madrid day hospital center, recorded legally and as part of the ICT4Life project. To provide a wider spectrum of scenarios, the testing set also includes two video sequences recorded in a shop and two other video sequences recorded in a university laboratory.

It is important to note that the two columns of the ZedBoard’s times in Table 2 represent only CPU processing mode without FPGA HW acceleration. The algorithm used in this test is the same as the one launched on the personal computer and Raspberry Pi 3.

Table 2 shows the time measurements of the experiment. It takes as reference the i7 times in milliseconds and presents the Rpi3 and ZedBoard measurements as a ratio, which expresses how much slower they are compared to the i7. Each measurement represents the time needed to process one frame. As a first consideration, the number of people per frame increases the complexity and the processing time (more people = more time needed to process the frame). The important consideration that emerges from the data is that the background subtraction is the slowest part of the algorithm; i.e., it is the module demanding more resources. The Kalman tracking, even if its complexity increases with a higher number of people per frame, has a minor impact on the total processing time. To guarantee real-time conditions, in this case, the processing time should not exceed 66.67 ms per frame.
(6)$$T_{FRAME}=BS+KT\leq 66.67ms$$

Only the personal computer could reach real time for video resolutions less than 1920 × 1080. The project aims to use an embedded low power device, but even Raspberry Pi 3 is not able to meet the timing requirements. For that reason a hardware acceleration is needed on the ZedBoard.

Having established the starting conditions, the parameter that is missing is the optimum number of frames to buffer in the RAM free zone. To find this parameter, some tests with the same video sequence and a different number (N) of frames buffered in memory must be carried out. The results are shown in Figure 10, wherein the blue curve represents the achieved frame rate and the latency is represented by the red curve; the principal vertical axis (on the left) shows the frames per second (FPS) speeds that the system can reach, the secondary vertical axis (on the right) shows the latency time in seconds and the horizontal axis shows the number of frames buffered in RAM. This simple experiment, which already uses the FPGA acceleration and a resolution of 480 × 480, makes it possible to choose the number of frames to buffer. In fact, the evolution of the blue curve shows that after six frames buffered in RAM, the system is able to surpass twenty FPS. The number of frames chosen was fifteen, which was the same frame rate of the input video, which represents a realistic compromise between memory usage and processing speed. The latency in this case represents the time that is needed by the system to process the video stream; in other words, how much time has passed since the first frames were introduced, processed and reached the output of the system. The latency observed was less than one second, which for all intensive purposes represents an acceptable value.

## 7. Results and Discussion

This section presents the results obtained by our prototype system, including the FPGA background subtraction module. The experimental setup is similar to the one described in the previous section, processing the same video sequences in order to maintain the same complexity (number of people, light condition, etc). However, as each video sequence has a different resolution, the video resolution of all the sequences have been normalized to generate different sets of video sequences. The selected resolutions are those available to the camera. Furthermore, we use two video format color spaces: RGB and gray-scale. Even if the original version of the algorithm is RGB, the gray-scale is added to evaluate the speed differences compared to RGB, as the footprint in memory of gray-scale video sequence is lower, enabling more pixels to be stored in FPGA memory simultaneously.

Table 3 and Table 4 show the execution time, in milliseconds, taken from different video sequences with different resolutions. Table 3 presents the results obtained using video sequences with a squared aspect ratio, while Table 4 presents the results with rectangular aspect ratio video sequences in order to compare them with state of the art systems.

Finally, Table 5 sums up the average timing performances of the different formats processed, aggregated by the color space used. In line with previous studies referenced above, the measurements are expressed in FPS. Table 5 has been converted from milliseconds to FPS to use the same unit of comparison depicted in Table 6.

The first observation of note can be made about the timing results shown in Table 3, Table 4 and Table 5, where we see that processing an RGB sequence introduces just a little overhead when compared to processing a gray-scale sequence, about 6% slower. The difference in overhead is primarily due to the memory copy operation, as the amount of memory required is triplicated. On the contrary, the overhead in the HW processing pipeline of the FPGA is negligible, as the HLS creates a similar internal circuit that could perform the background subtraction on the image pixels, regardless the number of components per pixel (one for gray-scale and three for RGB).

Another important observation that emerges from the results is that the number of people in the video sequences has a lesser impact on the processing time of the CPU+FPGA version than on the CPU-only version. In fact, Table 2 shows just CPU times, and you can see that the time processing increases when there are more people, as is expected. On the contrary, Table 3 and Table 4, which show the CPU+FPGA solution, show a negligible difference amongst all sequences. The generated background subtractor HW module has a data independent execution time, which is lower due to the HLS optimizations that generate a processing pipeline to implement parallelism.

The processing acceleration of the ZedBoard (CPU+FPGA) compared to ZedBoard (only CPU) is between 3.4X and 3.8X faster, while the processing acceleration of the ZedBoard (CPU+FPGA) compared to Raspberry Pi 3 is between 1.6 and 2X faster. Compared with the Intel i7 CPU the ZedBoard (CPU+FPGA) processing times are closer, but are still slower than Intel CPU. However, the ZedBoard, or other boards with CPU+FPGA SoC are embedded systems, which have inherent advantages in terms of cost, power consumption (that is less than 10% of the i7) and space.

Table 6 compares the performance of our system with other related systems, which lightly outperform our system. However, there are two factors not considered in the table: our proposal is a complete HW/SW chain that stores the data in main memory for other modules to use it, and the background subtractor module is automatically generated from the SW implementation. Our solution is more flexible, and its modification or integration in a SW processing chain is simpler and faster. The performance of the proposed system, with the bottlenecks included, is valid for real time Parkinson’s patient monitoring allowing its use in hospitals.

Table 7 compares only the performance of the background subtractor module, eliminating the effects of data input and output to main memory, as done in the related work. Our MoG hardware/software implementation achieves excellent performance marks.

Once the timing results are discussed, it is important to highlight another positive aspect: the reliability of the background subtractor implemented on the FPGA. This is important to analyze because of the approximations done, changing the variable types, as already mentioned and shown in Table 1. Figure 11 makes it possible to evaluate the results of the background subtractor module. It is arranged as follows: in the even rows the foreground masks are exposed, while in the odd rows the results of the tracking are shown. The images are also organized in columns where the gray-scale column represents the images processed with the FPGA using the gray-scale algorithm version, whereas the RGB column represents the images processed with the FPGA using the RGB algorithm version, and the reference column represents the images processed with the CPU using the original algorithm, to be used as a reference to compare and evaluate the FPGA processing capabilities. The analysis is done by capturing a frame for all the scenarios investigated (the hospital, the shop and the laboratory). It is important to note that in each scenario, the frame is the same for gray-scale, RGB and reference.

Evaluating the images, independently of the conversion from float to fixed, the people in the foreground masks are located exactly in the original position. Regarding their shapes, you can observe less sharpness in the figures, and in some zones the shape has been deleted. This could become an issue in the case that the Kalman tracker does not interpret the foreground correctly. In fact, in the Figure 11a,d the bounding box of the gray-scale algorithm has a minor area, and in Figure 11d there is a false positive, where in fact one person has been detected as two people. The same effect happens in Figure 11g,l,o,r. The RGB version displays in some cases a minor bounding box area, but does not generate false positives (detecting one person as two people), and in other cases, helps to reduce false negatives, as seen in Figure 11b,e, whereas in the reference images (Figure 11c,f) the tracking has marked the door, but there is no person at that position.

After the image comparison, a more complete analysis of the results was done using two performance indicators—*precision* and *recall*. In this case, *precision* represents the probability that a detected pedestrian is relevant, while the *recall* indicator is the probability that a relevant pedestrian is detected in a scene. The two indicators are defined as follows:(7)$$Precision=\frac{TP}{TP+FP}\;Recall=\frac{TP}{TP+FN}$$
where:*TP* (true positive) represents the bounding box estimated by the algorithm where a human is located in the sequence at specific time *t*.*FP* (false positive) represents the bounding box estimated by the algorithm where a human is not located in the sequence at specific time *t*.*FN* (false negative) represents the bounding box not estimated by the algorithm where a human is located in the sequence at specific time *t*.

Table 8 and Table 9 contain the respective results, the tracking precision and recall comparison for the algorithm, where i7 represents the personal computer and ZB stands for ZedBoard.

Finally, in Table 10 the FPGA resource utilization is depicted. As it is clear to observe, the RGB version of the algorithm needs more resources because each pixel is represented with three components, while the gray-scale is represented only with one component per pixel.

## 8. Conclusions

A complete tracking solution dedicated to biomedical purposes has been presented in this paper. The system has been tested in several scenarios: a shop, a laboratory and a day hospital, with special focus on the APM Day Hospital center in Madrid, where the system has been tested with Parkinson’s patients. The goal of this study was to create a solution that permits the tracking of patients while guaranteeing their privacy protection. To achieve this objective, a hybrid solution that uses the combination of hardware and software is proposed. The system runs on the novel Zynq platform that incorporates a CPU and an FPGA on the same chip. Moreover, a deep integration between hardware and software resources is proposed, which allows us to combine all the software libraries needed to track people, identify the most demanding parts of the software chain, and take advantage of the FPGA to accelerate it. The system developed could grant real time processing acquiring data from a commercial Zenith camera (@ 15 FPS), starting from the base resolution (192 × 192) up to 640 × 480. For applications that need higher resolutions, it is still possible to process in real time, but the Zenith camera should be configured to send a slower frame rate (8 FPS for 768 × 768 and 4 FPS for 1056 × 1056), as detailed in Table 5. This study also outlines analysis of the complete system reliability and testing of the two hardware implementations (that use RGB and gray-scale color spaces respectively) and the analysis of the algorithm that involves type conversion from float to fixed. The results achieved are compared with the reference results obtained by processing the same test sequences with the original version of the algorithm as the ground-truth. As detailed in the previous section, the reader can observe that the RGB version of the algorithm—although its efficiency is lower than that of the gray-scale version—obtains more accurate results. This is confirmed in Table 8 and Table 9 which show that the RGB implementation is the closest to the reference, even better in some cases. The RGB version is therefore the solution chosen for this project because it represents the best compromise between processing speed and accuracy.

Furthermore, the system presented is developed on a ZedBoard, an embedded system that allows real time processing, and additionally, it is a low-power consumption system. In fact, the board power consumption is lower than 20 W which represents a great savings in energy consumption when compared to a PC that will always have a much higher power demand (minimum 300 W).

Our future studies focus on the possibility of chip integration with the camera; in fact, many cameras already do this, streamlining the ability to acquire and process the data in real time using FPGAs. It is possible to integrate our background subtracion IP directly in the FPGA that acquires the data to provide a parallel processed streaming that could be analyzed with an external tracker using less resources. Another option would be to construct the entire Zynq chip inside the camera. This would allow the central control server to receive already-processed data with the tracking information from the camera. This would represent a great improvement looked at from an energy efficiency aspect, because the single chip power demand with the entire application is around 2 W. Additionally, a control server that previously could only handle a limited number of cameras, thanks to our solution would be able to control more cameras and cover a larger physical area. This also represents a great efficiency improvement in terms of energy savings.

Another future development foreseen in this study is the update to a newer and faster Zynq board. The bottleneck in the ZedBoard is represented in the RAM that operates at 533 MHz and has a low bandwidth and low data transfer rate. The entire system could easily be ported to a Zynq board with higher performance, which would enable a faster CPU and a higher bandwidth RAM with a higher data transfer rate. Thus, the system could deal with higher image resolutions, further improving the overall efficiency.

## Figures and Tables

**Figure 1 sensors-20-03189-f001:**
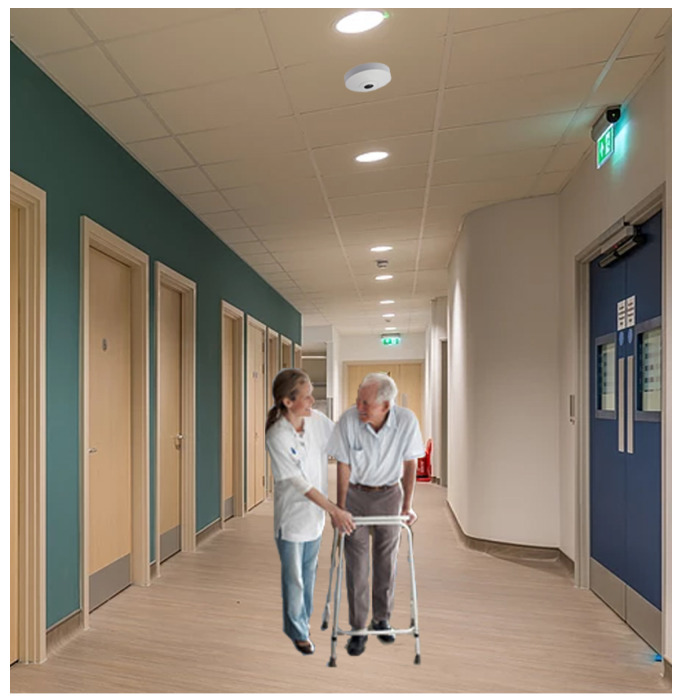
Typical application scenario.

**Figure 2 sensors-20-03189-f002:**
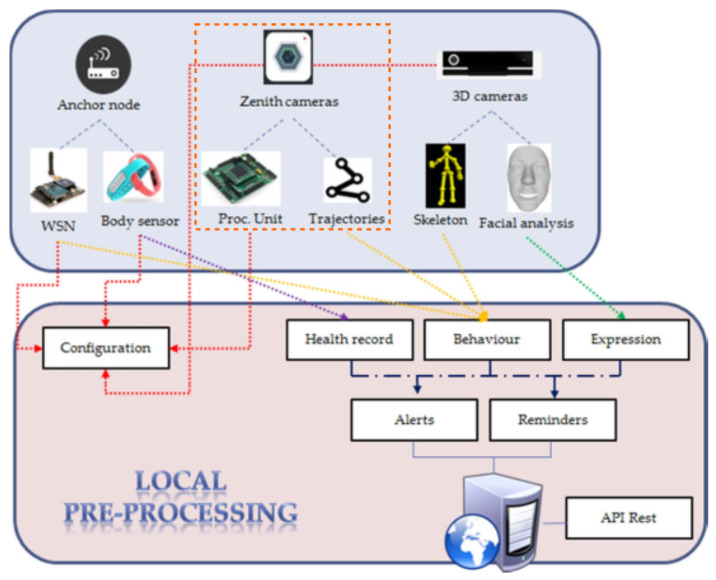
ICT4Life project structure from [14].

**Figure 3 sensors-20-03189-f003:**
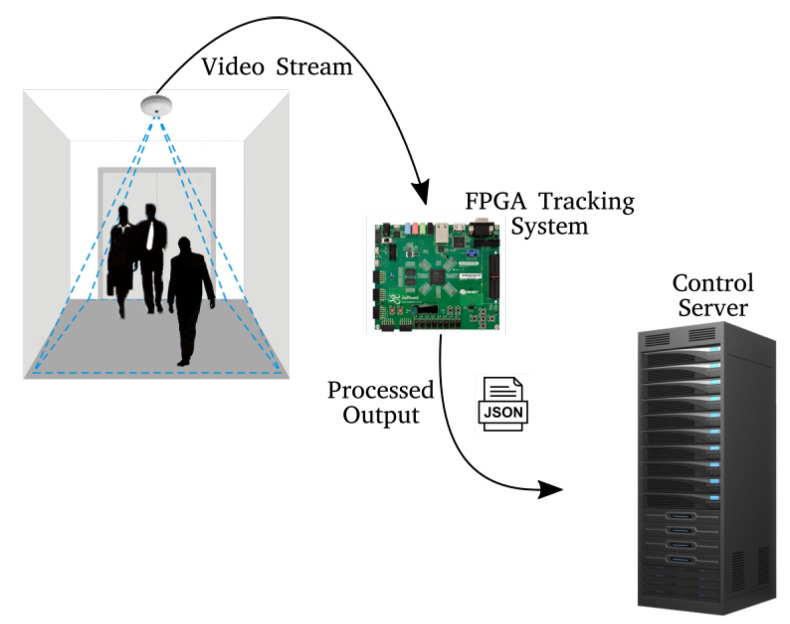
System elements considered: the Zenith camera (source), the ZedBoard (processing system) and control server (sink).

**Figure 4 sensors-20-03189-f004:**
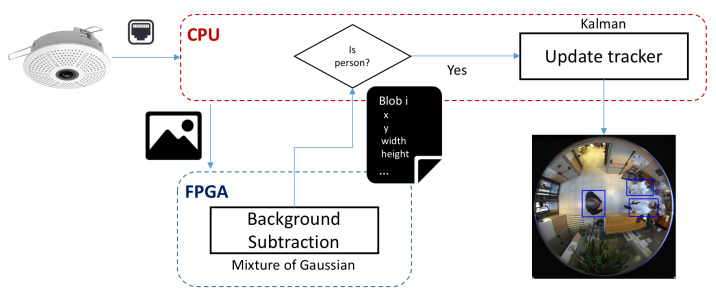
Workflow of the global tracking algorithm.

**Figure 5 sensors-20-03189-f005:**
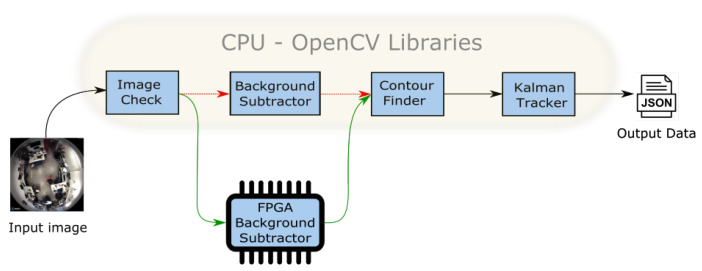
Software processing workflow to be modified and implemented into the FPGA.

**Figure 6 sensors-20-03189-f006:**
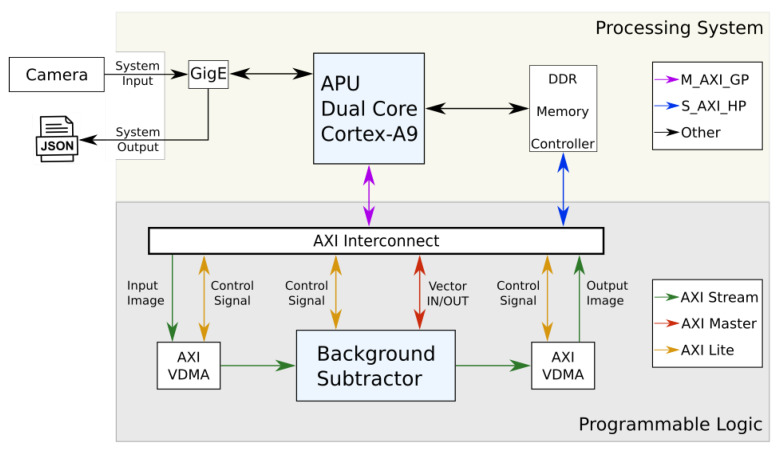
Zynq structure.

**Figure 7 sensors-20-03189-f007:**
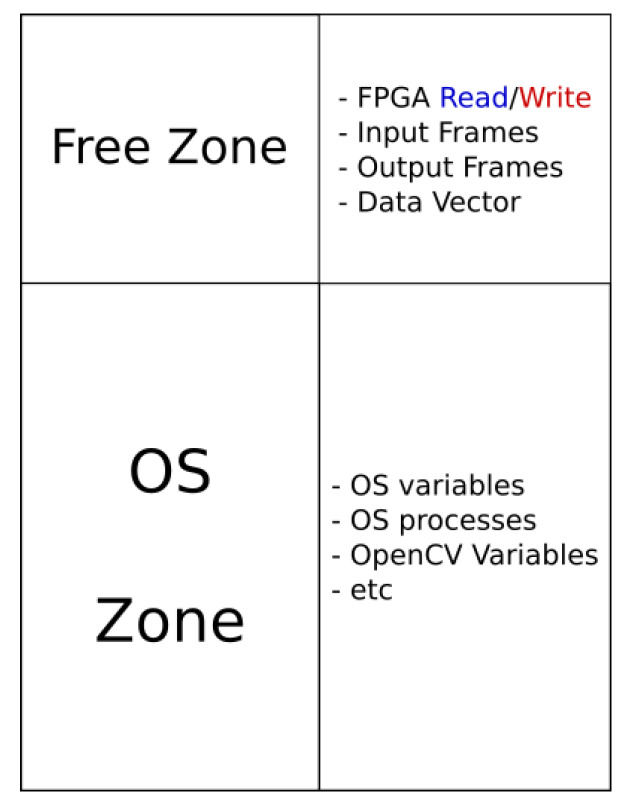
RAM distribution and utilization on the ZedBoard.

**Figure 8 sensors-20-03189-f008:**
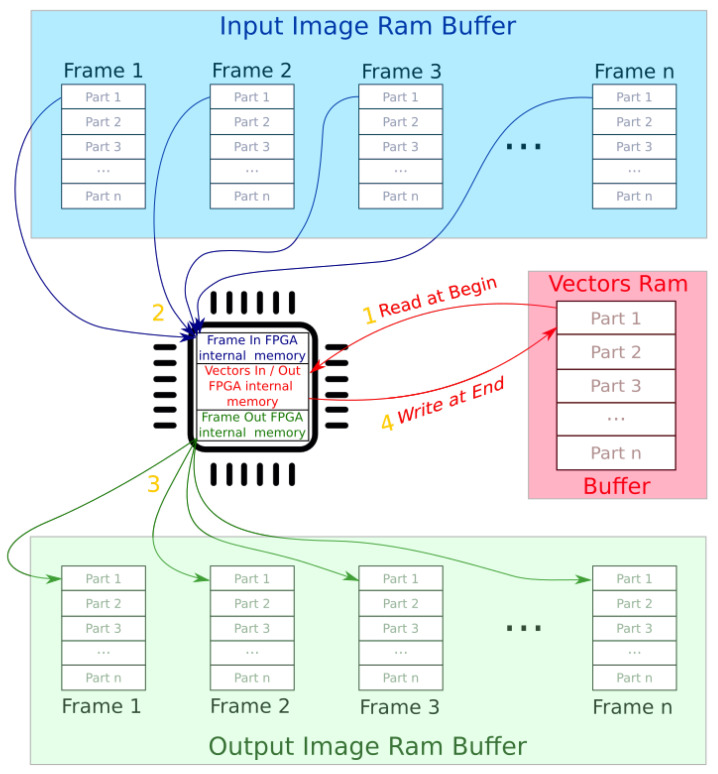
Frame buffering strategy used to move data.

**Figure 9 sensors-20-03189-f009:**
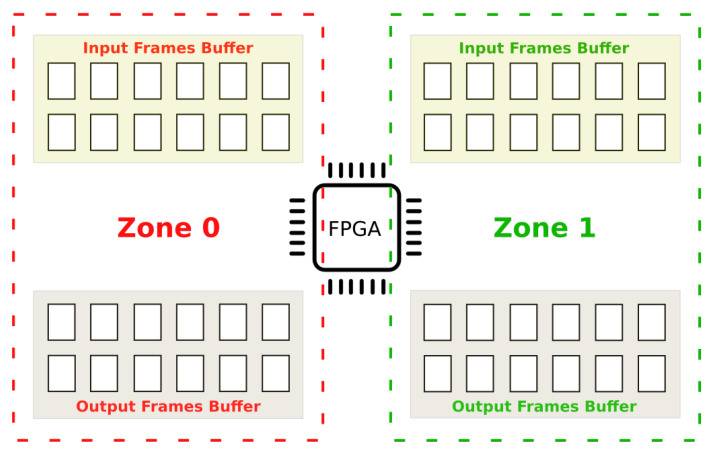
Multi-threading Frame buffering.

**Figure 10 sensors-20-03189-f010:**
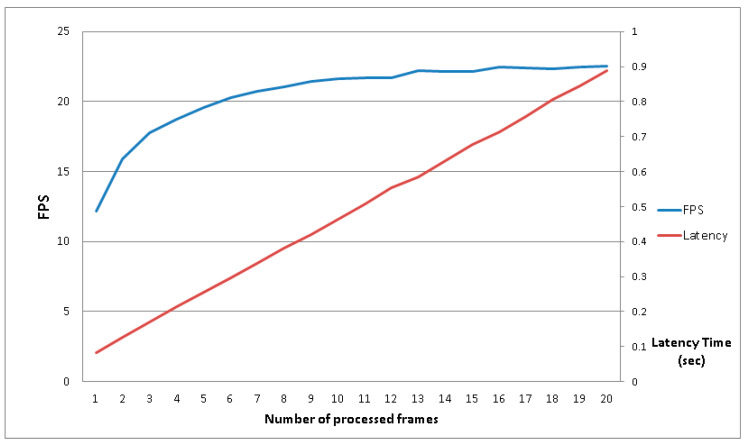
Evolution of the number of frames to buffer in RAM.

**Figure 11 sensors-20-03189-f011:**
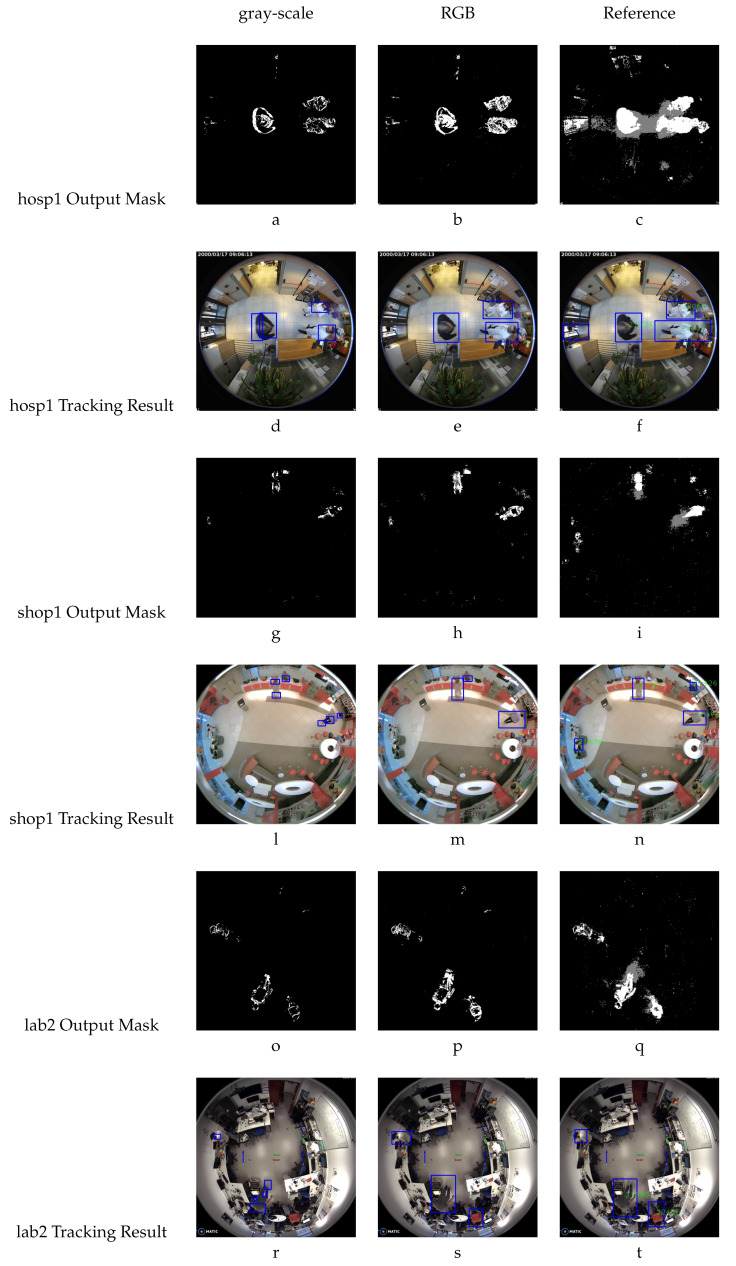
Images comparison (**a**–**t**).

**Table 1 sensors-20-03189-t001:** Data type conversion of the main variables.

Variable	Original Type	Size (in Bit)	FPGA Type	Size (in Bit)
Image In	unsigned char	8	unsigned char	8
Image Out	unsigned char	8	unsigned char	8
modeUsed[]	unsigned char	8	unsigned char	8
Variance[]	float	32	fixed	16
Weight[]	float	32	fixed	16
Mean[]	float	32	fixed	16

**Table 2 sensors-20-03189-t002:** Time measurements (in ms) to process one frame depending on the video sequence and the resolution for the i7. Rpi3 and ZedB are presented in i7 slower ratio. BS stands for background subtraction, KT stands for Kalman Tracking, i7 represents the personal computer, Rpi3 stands for Raspberry pi 3 and ZedB stands for ZedBoard.

Sequence	Resolution	Average Num of People/Frame	i7 BS (ms)	i7 KT (ms)	Rpi3 BS	Rpi3 KT	ZedB BS	ZedB KT
shop1.mp4	640 × 480	3.5	13.3	0.04	8.84	12.5	16.2	16.5
shop2.mp4	640 × 480	8	13.6	0.08	8.87	14.5	16.6	20.75
hosp1.mp4	1056 × 1056	4.3	48.9	0.045	8.02	11.11	17,2	14.88
hosp2.mp4	1056 × 1056	3	48.4	0.051	7.6	8.82	17.14	15.5
hosp3.mp4	1056 × 1056	3.3	48.5	0.042	7.56	9.3	17.13	15.48
hosp4.mp4	1056 × 1056	2.3	49.3	0.049	5.62	7.75	17.68	15.10
hosp5.mp4	1056 × 1056	5	48.4	0.056	7.8	6.96	16.98	10
lab1.mp4	854 × 480	2	17.5	0.038	8.82	12.37	16.69	17.37
lab2.mp4	1920 × 1080	3	88.7	0.05	8.93	12.2	17.03	15.4

**Table 3 sensors-20-03189-t003:** Time (in ms) to process the background subtraction on one frame depending on the video sequence and the resolution (square aspect ratio), using the ZedBoard (CPU+FPGA).

Sequence	480 × 480	480 × 480	512 × 512	512 × 512	768 × 768	768 × 768	1056 × 1056	1056 × 1056
	RGB (ms)	Gray (ms)	RGB (ms)	Gray (ms)	RGB (ms)	Gray (ms)	RGB (ms)	Gray (ms)
shop1.mp4	45.8	43.05	52.9	48.9	121.3	115.9	234.9	219.9
shop2.mp4	45.1	43.6	52.2	50.6	121.5	116.5	237.5	221.9
hosp1.mp4	47.4	44.6	53.5	49.05	123.9	123.9	242.2	228.7
hosp2.mp4	46.9	45.1	51.4	50	125.1	117.2	243.3	227.3
hosp3.mp4	47.2	44.6	52.5	51.8	124.3	124.3	245.8	229.9
hosp4.mp4	48.5	45.4	53.8	49.8	125.8	119.1	244.9	230.4
hosp5.mp4	46.5	43.8	53.5	50.5	124.2	117.9	242.9	229.8
lab1.mp4	44.6	43.3	50.9	49.2	121.9	115.2	233.2	222.2
lab2.mp4	44.9	43.2	50.3	48.8	118.7	115.4	232.1	219.9

**Table 4 sensors-20-03189-t004:** Time (in ms) to process the background subtraction on one frame depending on the video sequence and the resolution (rectangle aspect ratio), using the ZedBoard (CPU+FPGA).

Sequence	160 × 120	160 × 120	320 × 240	320 × 240	640 × 480	640 × 480
	RGB (ms)	Gray (ms)	RGB (ms)	Gray (ms)	RGB (ms)	Gray (ms)
shop1.mp4	3.9	3.6	15.3	13.7	62.2	59.04
shop2.mp4	4.1	3.8	15.5	13.9	63.6	59.7

**Table 5 sensors-20-03189-t005:** System average time measures in FPS, using the ZedBoard (CPU+FPGA).

Sequence	160 × 120	320 × 240	480 × 480	512 × 512	640 × 480	768 × 768	1056 × 1056
	FPS	FPS	FPS	FPS	FPS	FPS	FPS
RGB	250	64.93	21.58	19.11	15.9	8.13	4.17
Grayscale	277.8	72.46	22.7	20.07	16.84	8.45	4.43

**Table 6 sensors-20-03189-t006:** Main characteristics of literature tracking systems and differences with related work. NS stands for not specified, Eth stands for Ethernet, BGS stands for background subtractor.

Work	Hardware	Connectivity	Resolutions	Frame Rate	Power Consumption	Remarks
		IN/OUT		(FPS)	(W)	
Chen et al. 2017 [6]	Zynq 7020	Eth/NS	640 × 480	30	1.82	In/Out times not considered
Guo et al. 2017 [7]	Zynq 7020	PMOD/NS	320 × 240 resized to 160 × 120	89.2	2.99	Out system times not specified
Cocorullo et al. 2019 [8]	Zynq 7020 Virtex 6	NS	HD resized to 160 × 128 and 320 × 240	up to 74	NS	In/Out times not considered and not specified
Sajjanar et al. 2016 [10]	Zynq 7020	PMOD/VGA	640 × 480	81.57	NS	The chain process only the BGS
Zhang et al. 2018 [11]	Zynq 7020	NS	VOT2015 dataset	18.6	1.284	In/Out times not considered
Liu et al. 2015 [12]	Zynq 7020	NS/HDMI	1137 × 686	1.30	NS	In system times not specified
Cho et al. 2019 [13]	Virtex 5	NS/HDMI	640 × 480	30	NS	In system times not specified
Our Work	Zynq 7020	Eth/Eth	up to 1056 × 1056 see Table 5	up to 277.8 see Table 5	2.025 Gray 2.297 RGB	Complete chain analyzed, In/Out times considered

**Table 7 sensors-20-03189-t007:** FPGA mixture of Gaussian performance comparison.

Previous Works			Our Work			
Work	Real Resized Resolution	FPS	Resolution	FPS Complete Chain	FPS MoG Only FPGA	Remarks
Chen et al. 2017 [6]	640 × 480	30	640 × 480	15.9 (RGB) 16.84 (Gray)	162	I/O data exchange associated times are not considered
Guo et al. 2017 [7]	160 × 120	89.2	160 × 120	250 (RGB) 277.8 (Gray)	2600	The Input is resized, the Output data exchange associated times are not specified
Cocorullo et al. 2019 [8]	160 × 128 and 320 × 240	74	160 × 120 320 × 240	250 (RGB) 277.8 (Gray) 64.93(RGB) 72.46(Gray)	2438 650	The Input is resized, I/O data exchange associated times are not considered, the I/O interfaces are not specify
Sajjanar et al. 2016 [10]	640 × 480	81.57	640 × 480	15.9 (RGB) 16.84 (Gray)	162	The complete chain is specified, but no Tracking is performed, only MoG
Zhang et al. 2018 [11]	from 320 × 180 to 1280 × 720	18.6	320 × 240 1056 × 1056	64.93 (RGB) 72.46 (Gray) 4.17(RGB) 4.43 (Gray)	650 44.8	I/O data exchange associated times are not considered
Liu et al. 2015 [12]	1137686	1.30	1056 × 1056	4.17 (RGB) 4.43 (Gray)	44.8	The Input data exchange associated times are not specified
Cho et al. 2019 [13]	640 × 480	30	640 × 480	15.9 (RGB) 16.84 (Gray)	162	Only FPGA implementation, the time associated to input interface is not specified

**Table 8 sensors-20-03189-t008:** Tracking algorithm results comparison (represented in percentages).

Sequence	TP (%)	FP (%)	FN (%)	TP (%)	FP (%)	FN (%)	TP (%)	FP (%)	FN (%)
	i7 RGB	i7 RGB	i7 RGB	ZB Gray	ZB Gray	ZB Gray	ZB RGB	ZB RGB	ZB RGB
hosp1	81.78	9.98	4.23	48.94	38.80	12.26	72.01	18.13	9.86
hosp2	69.78	14.10	16.11	40.87	31.01	28.12	66	15	19
shop1	84.87	4.52	10.61	54.28	32.21	13.51	73.57	15.80	10.63
shop2	60.23	22.59	17.18	22.62	7.60	69.78	39.91	6.04	54.05
lab1	87.82	2.78	9.39	46.67	34.36	18.97	79.83	8.41	11.76
lab2	74.01	12.77	13.21	39.66	34.18	26.16	67.41	11.24	21.35

**Table 9 sensors-20-03189-t009:** Comparison of precision and recall for the tracking algorithm.

Sequence	Precision (%)	Recall (%)	Precision (%)	Recall (%)	Precision (%)	Recall (%)
	i7 RGB	i7 RGB	ZB Gray	ZB Gray	ZB RGB	ZB RGB
hosp1	89.58	93.22	55.78	79.97	79.89	87.95
hosp2	83.19	88.89	56.85	59.24	81.48	77.65
shop1	94.94	89.89	62.76	80.08	82.32	87.38
shop2	72.72	77.81	74.84	24.48	86.86	42.48
lab1	96.92	90.33	57.59	71.09	90.48	87.16
lab2	85.27	84.91	53.71	60.26	85.71	75.95

**Table 10 sensors-20-03189-t010:** FPGA resources utilization (in number of elements).

Resource	Available	Utilization	Utilization (%)	Utilization	Utilization (%)
		RGB	RGB	Gray	Gray
LUT	53,200	24,170	45.43	6892	12.95
LUTRAM	17,400	11,589	66.60	328	1.89
FF	106,400	11,588	10.89	9397	8.83
BRAM	140	135	96.43	111	79.29
DSP	220	39	17.73	24	10.91
BUFG	32	1	3.13	1	3.13

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
