# Peer review of "An FPGA Based Tracking Implementation for Parkinson’s Patients"

_sensors, 2020, doi:10.3390/s20113189_

Round 1
Reviewer 1 Report
The paper presents an FPGA implementation of a tracking system dedicated to a day hospital. It uses a Xilinx Zynq device that combines FPGA logic with CPU cores. The authors used a tracking method based on a background subtraction algorithm (Gaussian Mixture Model - GMM) combined with a Kalman filter. The initial implementation was done in software using a popular OpenCV library. The performed computations were analyzed and it was found out, that the background subtraction calculations were the most time-consuming. Hence that part should be accelerated in the FPGA logic. The conversion from a software approach to a hardware implementation was done using HLS (High-level synthesis) tool from Xilinx. The results were compared with a software implementation on a PC and a RaspberryPi board. With the proposed hardware implementation authors were able to speed up the algorithm compared to purely software implementation on RPi or CPU's of Zedboard. However, the cost was the reduced quality (reduction in Precision and Recall). Even with hardware acceleration, the solution was much slower than the PC based implementation.
It is worth noting, that authors present a working solution that was tested in a real-life scenario, not just an idea on how to help patients. However, in my opinion, the proposed solution is lacking novelty. The GMM algorithm is well established and known. There are already FPGA implementations described. The presented system is more of a combination of technical achievements (combining Linux, programmable logic, camera with the rest of the system) and the obtained results do not fully convince, that this approach is the right one. The speed-up is modest, while the quality of results is worse.
While the introduction mentions a lot of papers, in the result section there are no references to other implementations of GMM in FPGA. In my opinion, this is a must-have, especially that some papers are claiming real-time performance for FHD images ("ASIC and FPGA Implementation of the Gaussian Mixture Model Algorithm for Real-Time Segmentation of High Definition Video") or over 200 times faster than a 6-core Xeon ("A Fully-Pipelined Hardware Design for Gaussian Mixture Models"). Without that there are concerns about the most important part of this paper. A comparison with the existing solutions described is literature is a must, otherwise, it is difficult to judge if the proposed solution is good or not.
Things that could be improved:
- add comparison with more recent hardware - i7-4790 is from Q2'14. New processors could offer similar performance while using a lot less power. This is important, as there is a claim that Zedboard consumes only 10% of PC power.
- add comparison with a more powerful ARM board - maybe use a popular Nvidia Jetson? That will also allow for a comparison with the GPU accelerated algorithm. Also, Jetson's power usage will be more in line with the Zedbaord one.
- it is worth considering simpler background subtraction algorithms with additional post-processing. Those translate more efficiently to the FPGA logic. As far as I know, this is a promising approach. It allows creating a fully pipelined processing systems, which are extremely fast. The lower accuracy of simpler methods will be compensated by the additional post-processing operations (filtering, morphological operations), which are very "cheap" on FPGA, while are quite costly on PC.
- move data (images) reading directly to programmable logic. That way one unnecessary transfer of data will be removed.
- result presentation:
* beside a raw time show how much something is faster / slower. That is in table 2 write only i7 time in ms and present others as a ration of how much slower they are. So for shop1.mp4, it will be:
i7 BS - 13.3 9 [ms], Rpi3 BS - 8,84 [no unit], ZedB BS - 16,20 [no unit]
That way, the reader knows, that the RPi3 is over 8 times slower and Zedboard is over 16 times slower. If the reader wants time, he can easily calculate it. Using this approach also allows seeing if those ratios are the same for different resolutions.
* Tables 4 and 5 should be changed in a way, that they are more in line with Table 3. Right now it is difficult to compare them. One is showing processing time in ms, while the others present a performance measured in fps.
Other things:
- some parts, like the description of the platform (lines 141 - 161), are not needed. The other sensors, Kinect, beacons were not used in this work. Section 3.1 Design Methodology describes the basics of the HLS tool. I would prefer more details about the authors' implementation or more detailed results section instead of such a general and long introduction.
- concerns regarding the privacy of the patients are mentioned in the paper a few times. Some information is given about saving the debug images to a custom file, but details are lacking. It is said that the patient faces are not possible to be seen. Are there algorithms that detect the faces and blurs them? How is the file saved? Is it just a binary file without a header or is it secured by some kind of encryption?
- some editing needed - eg. Table 5 - Gray Scale -> Grayscale
To sum up - the paper presents a technical achievement but is lacking in scientific results and comparison to existing solutions.
Reviewer 2 Report
sensors-737903-peer-review-v1
An FPGA based tracking implementation for Parkinson Patients
Contribution: computer vision algorithm for segmentation to improve background subtraction of frames previously characterized by a Gaussian mixture model, implemented on an FPGA to provide a low-cost and real-time processing.
The contribution is not clear as it doesn’t strengthen the importance of tracking and Parkinson patients' subject. It should be important to mention that. The importance of trajectory and what is expected in the particular case of Parkinson?
The main contribution of this paper is the design of an embedded system that uses OpenCV
115 Libraries to track patients with Parkinson disease, tested in the day hospital center of Asociación
116 Parkinson Madrid (APM)1. -- describe the tracking process and how is related to Parkinson. What is supposed to provide and why background subtraction is needed in the process. Use also references [19] [20].
The author should motivate the importance of background subtraction before going to the state of art. And put this into context, Parkinson. After describing the previous work, it could be interesting to select some proposals interesting to compare (you must decide the parameters, capabilities, and requirements to use) by using a table; this can be general (just video processing) or particular (biomedical, FPGA-SoC-SoM).
It is still not clear the algorithm’s needed execution steps and associated data, in particular, regarding the hardware implementation architecture (algorithm itself). It could be important to depict the algorithm structure as it will be implemented.
There are no comparisons with state of art hardware implementations, just with the software version of the algorithm. Why then a long state of the art without a motivation of the algorithm?
Questions/remarks:
but it is not possible to use the advantage of MoG, sending the foreground mask output video flow to the next stage (for example to perform human pedestrian tracking). -- ??
The authors made a remarkable work, but do not specify how the output data is provided. – is that so important?
whose sequences are not constant in terms of resolution -- ??
Other works only extract Background Subtraction, for example from HDMI IN to HDMI OUT or from network camera to VGA out interface. In this work, the FPGA performs the complete segmentation and provides the foreground(FG) mask as an input to the the following stage. --- with the exception of “complete” the difference compared to “other works only” is still not clear.
and a frame rate of 15 fps -- How is that related to Parkinson? is that enough?
The Background Subtractor block is implemented using MoG [24] – can you motivate the choice of MoG at the state of the art section? “
236 The algorithm chosen for HW acceleration is MoG because of its stability for background 237 subtraction, and a Kalman filter for the tracking because of its robustness based on a statistical 238 model.” – this should be said at the SoA, plus what about the other requirements: speed, quality, performance, HW size? What do you mean by stability? Kalman filter is slow, due to its size, what about that? Should show SW vs HW execution time on both.
Verification follows to check if the design is meeting the time requirements and if 222 it is performing as expected. These three steps (C++ Functional verification, High Level Synthesis and 223 RTL verification) could be iterated if the design needs improvements or do not satisfy the requirements. – comment optimization strategies
the solution proposed in [26] has been initially adopted – justify
The training set is updated 246 by adding new samples and discarding the old ones -- what training? Training of what?
For that aim a uniform kernel -- ?
This might give too sparse sampling of the interval T -- ??
it is possible to individuate two parts -- ??
input images from the AXI VDMA, and the vectors data from the AXI Master bus. – vector data?
to generate an optimize hardware solution. – typo
This 350 is not possible to implement on a FPGA because it has a different structure and Vivado HLS does not 351 support this framework -- ??
Verify equation (5):
aT mk w (4) (5)
0,024 0,000288 0,5 0,048 0,024144
This leads to the need of analyzing the quantity of data to transfer for each sub image, 424 because it slows down the entire process when a sub image with all the vectors is introduced and this 425 should be repeated several times (e.g. 20 times for 640x480 resolution). -- ??
times are measured in two main point 465 of the chain -- grammar
The algorithm used in 468 this test, it is the same -- gram
video resolutions less than 1920x1080 – gram
processing frame rate proficiency -- ??
latency is represented from the red curve -- gram
Frame 488 Par Second – gram
Taking as starting point the experiment conditions aforementioned, -- gram
but the 504 video test resolution of all sequences have been normalized, according to a more complete set of video 505 resolutions compatibles with the camera -- ??
other two of an university laboratory – gram
the differences 520 between gray-scale and RGB processing decreases when the image resolution increases, because the 521 amount of data rises, and as a consequence the additional time needed for copy slows down the entire 522 process -- ??
the differences 520 between gray-scale and RGB processing decreases when the image resolution increases, -- show a table with the difference in latency between RGB and Gray
than only in the CPU -- ??
Table 4 shows the other aspect ratio formats available for comparison with state 514 of art. – where is that comparison?
Reviewer 3 Report
In my opinion this is a very good paper. Clearly written and properly organized. The proposed technique is proved by an experimental research.
The only one (minor) problem concerns language aspects (for example "a FPGA"), thus I strongly recommend solid English verification (preferably by a native).
Round 2
Reviewer 1 Report
The provided changes improve the quality of the paper but did not address my main concern - low novelty and proper comparison with available approaches.
I would recommend canceling the submission, finishing the work, that authors said are currently doing or planning to do ("Additional comparison and the integration of other functions for sure may enrich the work and are going to be considered for the upcoming works.") and resubmitting. That way, the article will be even better and could bring more interest and citations. I understand that these comparisons are not easy and require a lot of work. However, without, I am not convinced about what new knowledge this paper brings. I would like to once again stress out, that I am impressed and thankful for a working solution that helps patients with Parkinson's disease, but this is a technical achievement.
The added comparison to other FPGA implementations of the MoG algorithm shows that this implementation is slower, hence is choosing the HLS tools the right approach?
In summary, I appreciate the changes, but, as previously stated, without many improvements it is not worth acceptance. I do not think those can be done in a week or two, especially considering the current situation, so in my opinion, the paper should be resubmitted at the later date.
Reviewer 2 Report
sensors-737903-peer-review-v1
An FPGA based tracking implementation for Parkinson Patients
The requested changes are made, however, the document becomes too verbose. Several paragraphs are repeated around the document and a full reading should be done to make the text more concise.
Remarks:
An example of a biomedical device made on a FPGA is showed in [4], where a 3D ultrasound
39 sensing device has been developed. The proposed sensor is able to perform real-time 2D and 3D
40 complete ultrasound reconstruction with a power consumption of around 6W. The platform enables
41 telesonography by permitting the exploitation of ultrasound diagnosis by any untrained operator;
42 even in remote rural areas, underdeveloped regions, and rescue scenarios. The diagnostic results are
43 very close to a standard telesonography device, which is normally a bulky, high power consumption
44 and expensive machine available only in hospitals. – not fundamental or in the context of the actual work
The authors work is remarkable, however, they do not
56 specify how the output data was provided. – explain why is so important – not just how but also output format
When required, Xilinx Vivado HLS, which offers the possibility of designing and 222 modelling in C++, and can be used to synthesize a submodule in the FPGA section of the SoC. – rephrase
The module consuming more resources is selected, optimized 232 in software and changes in hardware if required. – rephrase
the change is highlighted by the arrows color. In red, the slow path,
253 and in green, the accelerated path with the FPGA implementation. -- ??
